# Adhesive Cements That Bond Soft Tissue Ex Vivo

**DOI:** 10.3390/ma12152473

**Published:** 2019-08-03

**Authors:** Xiuwen Li, Michael Pujari-Palmer, David Wenner, Philip Procter, Gerard Insley, Håkan Engqvist

**Affiliations:** 1Applied Material Science, Department of Engineering, Uppsala University, 75121 Uppsala, Sweden; 2GPBio Ltd., Unit 4D, Western Business Park, Shannon, V14 RW92 Co. Clare, Ireland

**Keywords:** tissue adhesive, phosphoserine, phosphoserine modified cement, biomaterial, bioceramic, lap shear, bone cement, silicate, calcium phosphate, self-setting

## Abstract

The aim of the present study was to evaluate the soft tissue bond strength of a newly developed, monomeric, biomimetic, tissue adhesive called phosphoserine modified cement (PMC). Two types of PMCs were evaluated using lap shear strength (LSS) testing, on porcine skin: a calcium metasilicate (CS1), and alpha tricalcium phosphate (αTCP) PMC. CS1 PCM bonded strongly to skin, reaching a peak LSS of 84, 132, and 154 KPa after curing for 0.5, 1.5, and 4 h, respectively. Cyanoacrylate and fibrin glues reached an LSS of 207 kPa and 33 kPa, respectively. αTCP PMCs reached a final LSS of ≈110 kPa. In soft tissues, stronger bond strengths were obtained with αTCP PMCs containing large amounts of amino acid (70–90 mol%), in contrast to prior studies in calcified tissues (30–50 mol%). When αTCP particle size was reduced by wet milling, and for CS1 PMCs, the strongest bonding was obtained with mole ratios of 30–50% phosphoserine. While PM-CPCs behave like stiff ceramics after setting, they bond to soft tissues, and warrant further investigation as tissue adhesives, particularly at the interface between hard and soft tissues.

## 1. Introduction

Skin is the largest organ and the first protective barrier against environmental insult. Small, simple skin lacerations can heal rapidly without intervention. However, if the injury is large, irregularly shaped, or deep, healing may be delayed. In cases of large and complex lacerations, the current standard of treatment is sutures, or staples. Staples are rapid and easy to apply, and sutures can effectively close irregular and large wounds. However, both treatments have limitations: sutures are more expensive, requiring surgical time to employ and remove, while staples are only applicable for wounds with straight edges [1,2]. Both approaches also have difficulty repositioning the skin edges exactly, potentially leading to undesired scar formation [1,3], infection [4], or chronic irritation during healing [5].

Soft tissue adhesives are an attractive alternative to sutures and staples, with a number of comparative advantages. Tissue adhesives are easy to apply, cause minimal trauma, they allow tissues to heal without needing to remove the adhesive at a later point, produce lower rates of infection [6,7] and less morbidity compared to staples and sutures, reduce operating time, and can improve cosmetic outcomes (i.e., less scaring) [1,8,9]. Many different adhesives have been explored for closing [10,11,12], or reattaching, injured soft tissues, including adhesives based on acrylate “superglue” [13], fibrin [14], polysaccharide [15,16], protein [1,17,18], or organic acid [19,20]. The chemistry underlying these adhesives utilize direct crosslinking with the tissue surface, via L-dopa bonding, aldehyde condensation, or enzymatic crosslinking (fibrin) [13,21,22]. 

The clinical requirements for an effective sealant/adhesive include: a) completely bridging the two laceration edges by sealing the wound (sealant) and/ or directly bonding the two edges (adhesion), even for irregularly shaped, large wounds; b) holding together the wound edges under physiological, tensile loading; c) biocompatibility (histo- and cyto-compatible, hemocompatible, etc.); d) performance in wet environments, such as physiological fluids; and e) degradation without impeding healing or inflammation, while also maintaining adhesive strength long enough to facilitate native wound closure [23]. Many of the proposed adhesives utilize crosslinking initiators that may be toxic or unsuited for specific applications (i.e., photoinitiation [24], or aldehyde crosslinking, which may elicit an immune response [25]), and weaken rapidly in liquids [26]. For example, tissues repositioned with acrylate-based adhesives may heal better than with sutures [27], yet acrylate glues can elicit a fibrotic response, and weaken under wet-field conditions [21]. In comparison, fibrin glues exhibit excellent histocompatibility, but are mechanically much weaker (10–50 kPa [19,28]).

Phosphoserine modified cements (PMCs), including the non-adhesive cement formulations studied by Reinstorf et al. [29,30,31], have been investigated for hard tissue applications because: (a) many of the chemical and physical properties of cured/set PMC mirror those of the mineral phase of bone (i.e., strong compressive strength [31], presence of amorphous and crystalline calcium phosphate, etc.); (b) phosphoserine exerts a direct stimulatory effect on the proliferation and differentiation of osseous cell types, including osteoblasts [32,33,34,35,36], stem cells [36,37], and endothelial cells [32,33,34,35,36], and can increase the expression of bone morphogenetic protein (BMP-2) and runt-related transcription factor (RUNX2) [32,36]; and (c) can improve the physical properties, (i.e., high compressive strength [31], strong bonding to calcified tissue surfaces [38,39,40], etc.), and immune-/histo-compatibility, of cements in vivo (i.e., by accelerating resorption and replacement of cement with new bone, and increasing the numbers of recruited macrophages and osteoclasts) [35,41]. PMCs have, only recently, been investigated for use as tissue adhesives [38,39,40].

Interestingly, many of the non-collagenous proteins that produce fracture resistance (toughness) in bone are highly phosphorylated peptides that are also involved in load dissipation, as well as interfacial adhesion, between the organic and inorganic phases of bone [42]. Osteopontin is a non-collagenous component of the osteoid with large amounts of (>10% of total amino acids) phosphorylated serine residues [43] that increases the fracture toughness of bones [44,45,46]. Osteopontin also produces an adhesive effect at the atomic to nanoscale [29,47,48] by crosslinking via ionic bonding with calcium [42,47,49]. These weak nanoscale interactions [50] collectively amount to an improved fracture energy many times stronger than covalent bonding [47]. The complex physiochemical and biochemical properties of osteopontin and other phosphoproteins, which arise from the three-dimensional conformation/chemistry, can also be reproduced with simple amino acid monomers of phosphoserine. Strong, tacky adhesion to calcified tissues, similar to what has been reported by Fantner and Hansma et al. [47,48,50,51], and improved resorption and osseous cell differentiation [52], were obtained from cements containing phosphoserine monomers rather than entire phosphoprotein macromolecules like osteopontin or osteocalcin (i.e., comparable benefits were observed in osseous defects treated with Biocement D containing either osteocalcin or phosphoserine [52]).

Since PMCs can bond to both hard and soft tissues, PMCs may also be suitable for bonding soft and hard tissue interfaces (i.e., at the tendon/bone interface). Before approaching such a complex application, however, PMCs must first be optimized for bonding to soft tissue surfaces. The purpose of this study, therefore, was to determine whether PMC was suited for a preclinical test model of wound apposition under tensile loading (soft tissue lap shear strength testing (LSS)). The factors affecting PMC bonding to soft tissue surfaces (i.e., composition, particle size, etc.) were evaluated by LSS testing. Adult porcine skin, from the ear, was used in this study because its properties are similar to those of human tissue [22,53,54,55]. Finally, a new type of PMC, which contained calcium silicate (CaSiO_3_, calcium metasilicate, CS1) was compared to PMCs containing calcium phosphate (Ca_3_ (PO_4_)_2_, alpha tricalcium phosphate, αTCP) to evaluate how the chemical properties of the mineral phase affected the final strength of the adhesive.

## 2. Materials and Methods 

### 2.1. Materials

All materials were purchased from Sigma-Aldrich (AB Sigma-Aldrich, Stockholm, Sweden), unless otherwise indicated. Alpha tricalcium phosphate (αTCP, Ca_3_(PO_4_)_2_) was synthesized as described previously [39]. O-phospho-L-serine, referred to hereafter as phosphoserine, was purchased from Flamma SpA (>95%, Flamma SpA, Bergamo, Italy). Cyanoacrylate (Loctite) and universal grips (Cocraft spring clamps) were purchased from Clas Ohlson (Uppsala, Sweden). Porcine skin was obtained from Lövsta kött (Uppsala, Sweden), with approval from the Department of Agriculture. 

### 2.2. Synthesis and Milling

Alpha tricalcium phosphate (αTCP, Ca3(PO4)2) was synthesized by heating (Carbolyte oven CWF1300, AB Ninolab, Stockholm, Sweden) calcium carbonate and monocalcium phosphate anhydrous (MCPA) at a 2:1 molar ratio on a zirconia setter plate for 12 h, at 1450 °C [39]. After quenching in air, the αTCP powder (αTCP2.1) was dry-milled (Reitsch PM400, AB Ninolab, Stockholm, Sweden) in a 500 mL zirconia milling jar, at 300 rpm for 15 min, with 100 g of powder per 100 zirconia milling balls (10 mm diameter). αTCP was also milled using different milling conditions to produce smaller average particle-sized powders. To produce αTCP1.9, 100 g of powder was milled with 144 g of 5 mm diameter zirconia balls (≈360 balls), 15 mL isopropanol, at 300 rpm, for 15 min in total, in 1 min intervals, with 10 s break times and reverse rotation. To produce αTCP1.5, 100 g of powder was milled with 196.5 g of 2 mm diameter zirconia balls, 17.5 mL isopropanol, milling at 450 rpm, for 30 min in total at 1 min intervals, with 10 s break times and reverse rotation. Each wet milled powder was dried in atmosphere for 12 h to remove any residual isopropanol. Calcium metasilicate (wallastonite, hereafter referred to as CS1) was used as received from Sigma-Aldrich. O-phospho-L-serine, hereafter referred to as phosphoserine, (>95%, Flamma SpA, Bergamo, Italy) was used as received. Fibrin glue was prepared by dissolving 200 U of Thrombin (bovine, 31 units/mg) per 0.4 mL of 40 mM CaCl_2_; 40 mg of fibrin (bovine, <75% clottable protein) per 0.4 mL of water, and combined in equal volumes (20 µL total volume).

### 2.3. Materials Characterization

The particle size of each powder was analyzed using laser diffraction on 0.2 g of powder in 15 mL of isopropanol dispersed with sonication (Elmasonic S50R, Elma Schmidbauer, Singen, Germany) for 5 min. Approximately 7.5 mL was loaded (obscuration of 10–20%) into a Mastersizer 3000, with a hydroEV wet dispersion unit (Malvern Instruments Nordic AB, Uppsala, Sweden). Values of 1.63, 0.1, and 1.39 were used for particle refractive index, absorption index, and dispersant refractive index, respectively. Particle size analysis was conducted with software provided by the manufacturer (Mastersizer 3000, version 3.5). Throughout the manuscript, the different particle sizes of αTCP (1.5, 1.9, 2.1) refers to the average particle size (D0.5, surface %, in micrometers, Figure 2A) of the calcium salt portion of PMC.

Scanning electron microscopy (SEM) images were obtained with a Merlin field emission SEM (AB Carl Zeiss, Stockholm, Sweden), with an secondary electron in-lens detector, an acceleration voltage of 3 keV, and a 195 pA current, at a working distance of 5 mm. Samples were sputtered prior to SEM analysis with a (10-nm thick) coating of gold and palladium (Emitech SC7640, Quorum technologies, Kent, U.K.) at 2 kV for 40 s.

X-ray diffractograms (XRD) of powder samples were obtained on a Bruker D800 advance (Bruker Daltonics Scandinavia AB, Solna, Sweden), scanned with a step size of 0.03 degrees per step, from 3 to 60 degrees. The composition of each powder was determined using Rietveld refinement on Profex software (https://profex.doebelin.org) [56], with the following references for αTCP: PDF# 04-010-4348 αTCP, #01-074-0565 hydroxyapatite, #04-007-9734 CaO, #04-008-8714 βTCP). High quality structure files (.str) were not available for wallastonite (CS1), therefore the following reference patterns were manually compared to the diffractograms to identify additional phases present in CS1: α-wollastonite (αCS1, JCPDS 10-486, JCPDF 43-1460), β-wollastonite (βCS1, JCPDF 31-0300), dicalcium silicate (CS2 orthorhombic PDF#04-010-958, monoclinic “Larnite” #04-007-2687, and “hillebrandite” “CS2-H” #04-012-1668). These were used to identify peaks on the obtained diffractograms. No peaks corresponding to tricalcium silicate (CS3, monoclinic “haturite” #0-4014-9801, rhombohedral #04-011-1393, triclinic #04-012-3692), or silicon dioxide (i.e., cristobalite #39-1425, tridymite #42-1401, or quartz #46-1045), were observed. The following refinement parameters were used with Profex: Lorentzian size distribution of grain size (*b1*, *anisotropically*), microstrain (*k2*), and preferred orientation (*SPAR6*); other structural parameters were kept fixed, with a error of approximately 1% (phases present in concentrations below 1% cannot accurately be evaluated).

Specific surface areas were determined using nitrogen gas adsorption according to the Brunauer, Emmel, and Teller (BET) method on a Micrometrics ASAP 2020 (Micrometrics, Chemical Instruments AB, Stockholm, Sweden), with software supplied by the manufacturer and compared to values derived from laser diffraction based on surface density calculations.

### 2.4. Sample Preparation

Adhesive test samples were prepared by gluing together two strips of porcine skin with either an adhesive cement (PMC), cyanoacrylate (Loctite), or fibrin glue. Porcine skin (from ears) was detached from the underlying cartilage with a spatula, covered with PBS solution-soaked tissue, and stored at −20 °C for later testing. Immediately prior to testing, frozen skin was gently defrosted at room temperature and cut into rectangular strips (1 cm × 2 cm, Figure 1A) using a surgical scalpel. The average thickness of the obtained skin was ≈1.5 mm. Samples were prepared and tested as described in the American society for testing and materials (ASTM) standard F2258, unless otherwise indicated.

Adhesive cements (PMCs) were created by hand-mixing powders with a defined mole percentage of phosphoserine to αTCP, or phosphoserine to calcium metasilicate (i.e., a 50 mol% formulation contained 1 mole of phosphoserine powder for every 1 mole of αTCP). Deionized water containing a retardant (trisodium citrate, 1.3 M), or no retardant (NR)), was added to PMC as the liquid, and the mixture was hand-mixed with a spatula for 20 s. A liquid-to-powder ratio of 0.25 g mL^−1^ was used. Adhered skin samples (sized 1 cm × 2 cm) were prepared by applying a thin layer (0.25 g) of PM-CPC on the skin surface (overlap area 1 cm^2^). The adhered skin strips were held together with universal grips and stored in 100% humid sealed containers at 37 °C throughout the curing process (0.5–4 h). A comparable volume of Loctite (cyanoacrylate glue, ≈20 µL), or fibrin glue (10 µL thrombin, 10 µL fibrin solution), was used as reference test samples.

### 2.5. Mechanical Testing

Adhered skin strips were tested in lap shear. Samples were loaded to failure on a Shimadzu AGS-X mechanical testing machine (Shimadzu Europa Gmbh, Duisburg, Germany, Figure 1C) at a crosshead speed of 10 mm per minute, unless otherwise indicated. The resulting data was analyzed with Trapezium Lite software (version 1.0.1), provided by the manufacturer.

### 2.6. Statistical Analysis

Comparisons between two groups were evaluated with a student’s t-test. Comparisons between more than two groups were evaluated with a one-way ANOVA, with SPSS software (version 22), using Games–Howell post hoc analysis. * and ** indicate *p* < 0.05 and *p* < 0.01, respectively.

## 3. Results

### 3.1. Composition, Crystallinity, and Particle Size of PMC Precursors 

An overview of the testing process is shown in Figure 1. Prior to testing the particle size distribution of the starting materials (mineral precursor), αTCP and calcium metasilicate were evaluated using laser diffraction (Figure 2). The volume-, surface-, and number-weighted mean particle size that represented 50% (Figure 2A) or 90% (Figure 2B) of all measured particles, revealed large size differences between differently milled αTCP grains. A bimodal distribution was observed in αTCPs1.5–2.1 (Figure 2C), with comparable profiles below 1 μm, and differing distributions in the larger-sized particles (1–80 µm). The specific surface area values obtained from gas adsorption (BET) and surface weighted laser diffraction calculations (Figure 2D) were similar. Though αTCP1.9 appeared to be similar to CS1 with respect to surface area (1.81 vs. 1.481 m^2^ g^−1^) and surface weighted particle size (D(3,2) 4.07 vs. 4.11), the CS1 distribution was not bimodal, and no particles above 10 µm were detected in CS1 (Figure 2C).

SEM analysis confirmed the particle size analysis results, with increased milling producing an obvious decrease in mean αTCP particle size, though larger particles are still visible (Figure 3). The CS1 particle distribution appears (Figure 3D) visually similar to αTCP1.9 (Figure 3B). In all cases, the surface topography appeared relatively smooth and granular, with the exception of αTCP 1.5. The particle surface appeared significantly rougher and irregularly shaped (αTCP 1.5), partially due to the presence of smaller particles. 

Concurrent with the increased surface area and roughness of αTCP1.5, XRD peaks were also broadened (Figure 4), which suggest that the milling process created poorly crystalline/disordered regions within αTCP1.5 and αTCP1.9 particles. Peaks corresponding to common contaminating phases (hydroxyapatite, βTCP, etc.) were not obvious in the XRD diffractogram, though Rietveld analysis revealed that all αTCPs contained <2 wt% βTCP (PDF#04-008-8714). In CS1, powder peaks corresponded to both α- and β- forms of calcium metasilicate, and dicalcium silicate (Hillebrandite (CS2)).

### 3.2. Selection of Test Conditions

Prior to final testing, the optimal testing conditions were evaluated using lap shear tensile tests (Figure 5). Skin was isolated from the caudal (back side) and rostral side of the ear and stored separately. A direct comparison of the two skin sources demonstrated that they could be used interchangeably without producing different bond strengths with PMC2.1 (Figure 5A, *p* = 0.837). Calcium cements are typically tested at a crosshead speed of 1 mm per minute due to rapid crack propagation during brittle failure, which requires slower loading to measure the failure point accurately. Soft tissues are typically tested at between 10–30 mm per minute to prevent viscoelastic creep and fatigue from confounding the results. Therefore, three crosshead speeds were tested to evaluate the contributions from the tissue type and adhesive group to failure (Figure 5B). The average bond strength remained unchanged, regardless of crosshead speed for αTCP2.1 PMC, after curing for 90 min. 

Few studies have evaluated how the contact surface area affects the final LSS results. Since actual lacerations differ in size between patients, it is necessary to evaluate how a soft tissue adhesives’ LSS values would scale, depending on the size of the laceration. Three different specimen sizes were used, to approximate small- to moderate-sized lacerations, and the obtained strength was normalized against contact surface area. Smaller overlap surface areas tended to inflate the normalized LSS (Figure 5C), while only modest changes in normalized LSS occured with samples between 1 cm^2^ and 2 cm^2^. Standard test sample sizes of 1 cm × 2 cm long (1 cm^2^ overlap area) were used for all subsequent tests, as recommended by relevant test standards (ASTM F2258). 

Another potential confounding factor is the contribution from universal grips to maintain contact between the adhered surfaces while the adhesive cures. In a direct comparison between different samples that were tested after curing with, or without universal grips, there is a clear increase in bond strength from the applied pressure of the grips. Perhaps most importantly, since PMC cures as a thick paste, similar to calcium cements, then (a) the contribution from the grips was essentially linear (roughly, a 40–60% decrease in LSS when comparing gripped to ungripped PMC samples (Figure 5D)); and (b) cement that was not adhesive (αTCP) produced negligible LSS, confirming that the grips did not inflate the LSS values for thick pastes/cements (6 kPa, 0% group). Skin that was held together with grips and without adhesive (sham) exhibited a significant attractive force (50 kPa), confirming that for soft adhesives, the use of grips artificially increased the adhesive force (an artifact). 

### 3.3. Lap Shear Strength of PMCs

In the present study, the optimal ratio for CS1 PMC (Figure 6A, green line) was approximately 53%, while in αTCP2.1, the highest LSS strengths were obtained with PMCs containing 87% phosphoserine (orange, blue line). The higher pH of CS1 accelerated the acid–base reaction of PMCs, therefore a retardant was required to ensure a sufficient working time. The same retardant concentration (trisodium citrate 1.3 M) was used for all samples, except the “αTCP2.1 NR” group (Figure 6A, blue line). Since αTCP is less alkaline and cures slower than CS1 PMCs, a separate group was tested without retardant to determine whether the retardant affected the final LSS values. After curing for 90 min, the highest strength formulations of αTCP and CS1 PMCs reached 132 and 102 kPa, respectively. For reference, the observed LSS of cyanoacrylate and fibrin glue are shown in Figure 6A.

PMCs containing different particle sizes of αTCP produced significantly different LSSs (Figure 6B). αTCP1.5 PMC produced an LSS profile similar to CS1 PMC, and previous reports of PMC in hard tissue shear testing showed that the strongest LSS was obtained between 30–50% phosphoserine [39]. Intermediate particle sized αTCP1.9 PMCs appeared to combine elements of both large and smaller particles, remaining equally strong over all phosphoserine ratios. 

Considering the LSS profile, there appeared to be two regions of interest for further investigation: 53% (peak LSS for CS1 and αTCP1.5 PMC) and 87% (peak LSS for αTCP2.1 PMC) phosphoserine. PMC required approximately 1 h of curing time to exceed 50% of the final strength, and at least 4 h to exceed 90% of final bond strength. The LSS of each PMC was compared at the two regions of interest (53%, Figure 6C, and 87% phosphoserine, Figure 6D) at different curing times to determine whether the final strength differed between the different PMCs at 4 h as it did after 1.5 h of curing. 

At 53%, phosphoserine CS1 and αTCP1.5 PMCs behaved similarly, continuing to increase in bond strength between 1.5 and 4 h, and reaching a final LSS of 153 and 94 kPa, respectively. The LSS did not increase between 1.5 and 4 h for αTCP1.9 and αTCP2.1 PMCs (LSS 26–44 kPa). There appeared to be a relationship between particle size and final strength at 53% phosphoserine, where smaller particle sizes produced a greater LSS. At 87%, phosphoserine αTCP1.5 PMCs were equally strong compared to the LSS at 53% phosphoserine after 4 h of curing. CS1 PMCs produced a significantly lower LSS at 87% phosphoserine. All αTCP PMCs produced comparable LSS at 87% phosphoserine. Though αTCP PMCs were not, ultimately, as strong as CS1, it is likely that longer curing times would produce even stronger LSS values, and that these values may reflect the rate of curing, rather than the ultimate LSS.

## 4. Discussion

Phosphoserine-modified cements are tacky, adhesive pastes that bond to multiple tissue types, including calcified tissues, soft tissues, internal organs, and cartilage. Previous studies have shown that the ratio of phosphoserine to calcium salt should be approximately 30–50 mol% to obtain the strongest shear bond strength with calcified tissue surfaces. In the present study, αTCP PMCs with higher ratios of phosphoserine produced stronger adhesive strength to soft tissue. Interestingly, by reducing the particle size and increasing the surface area of αTCP PMCs, the adhesive properties changed: the soft tissue LSS of PMCs with smaller particle size was the strongest at 30–50 mol% (4-fold stronger than αTCP2.1 at 30%), similar to previous studies. This may be due to the dissolution rate, which varies with particle size and surface area, or directly due to the interaction surface area between the two reactants (αTCP and phosphoserine), or perhaps due to change in the modulus of the set adhesive. Stiffer moduli adhesives are known to produce higher bond forces, though perhaps PMCs with a higher phosphoserine content have a modulus more appropriate for bonding to soft tissue. Ongoing work is focused on addressing these questions.

The most significant finding was that, when bonded to skin, PMCs reached 40% of the humid LSS bond strength of cyanoacrylate within 30 min, and 47–60% (102 and 132 kPa for αTCP2.1 NR and CS1, respectively) within 90 min. The present PMCs were up to 44-fold stronger than fibrin glue, and up to 3-fold stronger than other mussel-inspired glues [1,5,19,26,28,38] after 90 min. Surprisingly, a thorough review of the literature reveals that there is no recommended minimal strength for a soft tissue (skin) adhesive. Since modified acrylates, such as Dermabond, yield clinical and cosmetic outcomes comparable to suturing [57], these adhesives appear to be suitable standards by which to compare the adhesive strength of PMCs. The typical range of adhesive bonding strength to skin for biomimetic and naturally derived adhesives is approximately 10–200 kPa [13,26]. In the present study, the adhesive strength of PMCs were on the high end of this scale.

In the present study, fibrin glue and cyanoacrylate were used as reference materials for comparing the adhesive strength of PMCs to soft tissue because: (a) these are the two most successful bonding agents (including use as a sealant or adhesive) used clinically; (b) the US food and drug administration (USFDA) has approved acrylate-based (i.e., Dermabond, produced by Johnson and Johnson, approved for topical closure of lacerations) and fibrin-based (i.e., ARTISS, produced by Baxter, approved for the adhesion of skin grafts in cases of burn injury) commercial products for use as soft tissue adhesives; (c) there is no similar *adhesive control* material (i.e., an adhesive cement) reported in the literature for use with soft tissue; and (d) similar materials that could be considered a “*material control*” (i.e., calcium phosphate cement using the same starting material, αTCP) display a negligible adhesive strength to soft materials. 

Another novel finding in the present study was that calcium metasilicate (Wallastonite) could produce a self-setting cement, in addition to strong soft-tissue adhesive strength during the curing phase, when combined with phosphoserine. This is the first published report of a wallastonite-based, self-setting cement. This is also the first report of an adhesive, or PMC, that uses wallastonite to create tissue adhesion. The bond strength of pristine αTCP cement to skin was negligible (6 kPa, 0% group, cured without grips), which confirms the observed adhesive bonding is a direct result of the inclusion of phosphoserine in PMCs. For hard tissues, a similar type of PMC was investigated by Kirillova et al., with the main difference to the present study being that Kirillova used tetracalcium phosphate (TTCP) rather than αTCP as the mineral portion. In our hands, PMCs containing TTCP reacted too quickly, partially due to the larger difference in pH between the acidic amino acid (phosphoserine- 3.5–4.5 pH range) and basic calcium salt (TTCP 9–11 pH range), and were difficult to handle and apply to soft tissue surfaces. Therefore, a direct comparison between TTCP and αTCP PMCs was not possible in this study. Additional studies are also needed to determine whether the biological and histological responses to both types of PMC are comparable. 

Interestingly, the type of PMC previously studied by Reinstorf et al., Mai et al., Offer et al., Schneider et al., and others [30,31,35,41,58,59] (Biocement D), contained collagen (2.5 wt%) as well as phosphoserine (1–10 wt%). When collagen was excluded from the cement, and less phosphoserine was used (1 wt%), the PMC resorption rate was not as pronounced as expected, though a *material* control group (Biocement D with collagen, or collagen and phosphoserine) was not included in that study [35]. When phosphoserine levels were higher (5–10 wt%) the cement was denser with less surface area [31], likely due to the liquefying effect of acidic phosphoserine, as has been described for citric acid and other cement additives [60,61]. As a cement becomes denser, less porous, or the surface area available for cell mediated degradation decreases, the degradation rate decreases. Therefore, one concern with using PMCs with greater amounts (>5 wt%) of phosphoserine is that the resorption profile may differ from what has been reported [30,31,35,41]. Recent in vivo studies on TTCP-based PMCs reported that PMC was largely degraded (<25% remaining) within 52 weeks in rabbits [38]. However, the healing and remodeling rate in smaller animals, including rabbits, are often more rapid than in humans. This may also explain the differences between prior PMC studies in rats and mini-pigs showing rapid resorption [52,58,59], while in larger animals (sheep [35]), which experience remodeling rates similar to humans, significantly less/slower resorption of PMCs was reported. Fortunately, PMCs are applied as thin layers (<0.5 mm thick) when used as an adhesive and are, therefore, more likely to degrade within a reasonable time frame. Future studies should address the differences between current and past types of PMCs and their respective resorption rates in an appropriate large animal model.

It should be noted that the present study was limited in cure time: a maximum cure time of 4 h was selected for the present study because the mechanical properties of skin samples deteriorated and may not have been reliable when incubated for longer time periods (i.e., 8 h or more) in warm, humid environments. Soft tissue also presents a challenge as the magnitude of differences observed between different groups was on the order of 25 kPa. Therefore, the present PMCs will be investigated further in hard tissues, where significantly larger forces are generated, in future studies.

## Figures and Tables

**Figure 1 materials-12-02473-f001:**
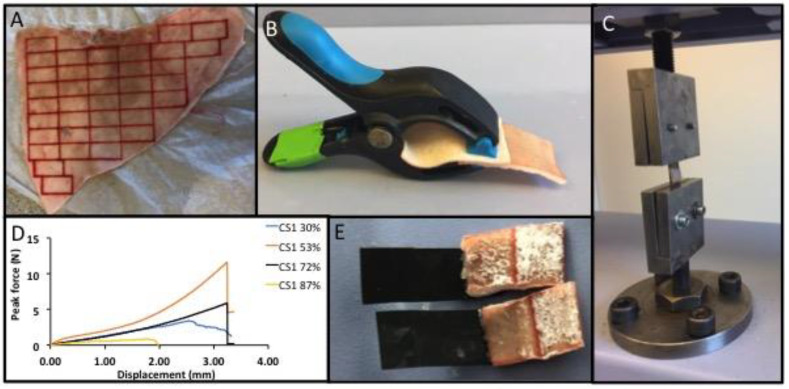
(**A**) Overview of the testing procedure and obtained data. Skin was removed from an entire ear, as a single piece, and subsequently sectioned into smaller strips 1 cm × 2 cm. (**B**) To ensure reproducible and comparable conditions each sample was held together during the curing stage using universal grips. (**C**) The lap shear testing setup. (**D**) Representative force/displacement curves for calcium metasilicate PMC (CS1), and the peak force per area was calculated for each sample. (**E**) Adhesive PMCs were applied to the skin surface as a viscous, tacky paste, with the failure surface.

**Figure 2 materials-12-02473-f002:**
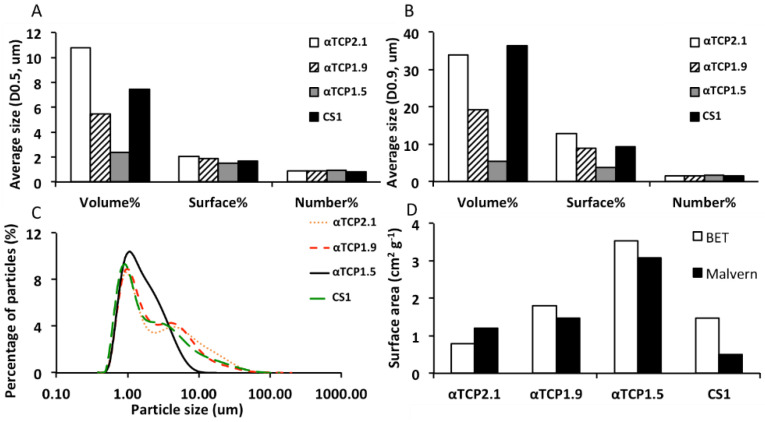
Particle size distribution of PMC precursors. The particle size limit of each PMC precursor (calcium salt) below which, based upon number, surface, and volume average calculations, (**A**) 50% (D0.5) or (**B**) 90% (D0.9) of all measured particles fall. (**C**) A distribution plot of all PMCs using surface average calculations. The surface area of each PMC precursor (**D**), determined by BET (white bars) or laser diffraction (black bars). Each αTCP (i.e., αTCP2.1) refers to the average particle size (D0.5, surface, in micrometers) of the calcium salt portion of PMC.

**Figure 3 materials-12-02473-f003:**
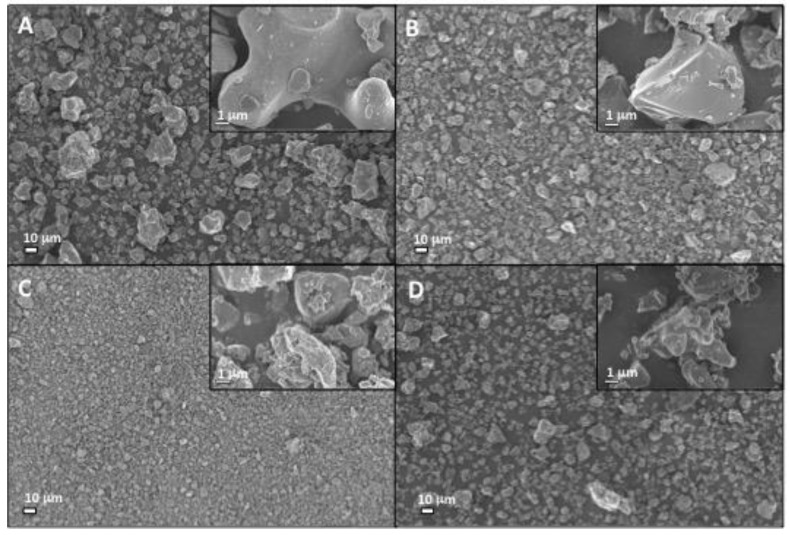
Secondary electron SEM images of precursor calcium salts: (**A**) αTCP2.1, (**B**) αTCP1.9, (**C**) αTCP1.5, and (**D**) CS1 particles are shown at 1000× magnification. Inset images in the upper right corner reveal the surface topography and fine structure of respective particles at approximately 25,000× magnification. Each number following the precursor name (i.e., αTCP2.1) refers to the average particle size (D0.5, surface, in micrometers) of the calcium salt portion of PMC. Scale bars represent a 10 µm distance, while inset scale bars represent a 1 µm distance.

**Figure 4 materials-12-02473-f004:**
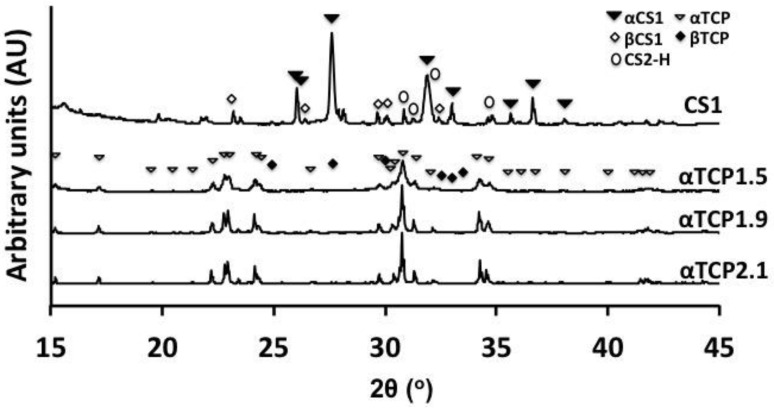
X-ray diffraction of calcium salt precursors used to make PMCs.

**Figure 5 materials-12-02473-f005:**
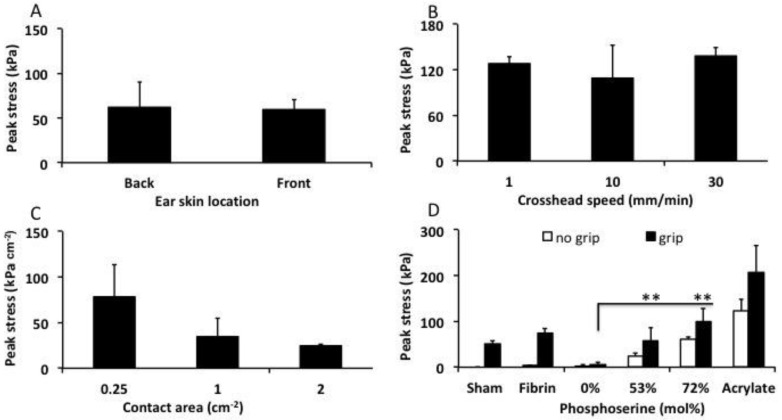
Lap shear strength of PMCs under varied testing conditions. (**A**) The bond strength of αTCP2.1 PMCs applied to porcine skin obtained from either the rostral (front) or caudal (back) portion of the ear. (**B**) Bond strength of αTCP2.1 PMCs tested at a crosshead speed of 1, 10, or 30 mm per minute. (**C**) Bond strength of αTCP2.1 PMCs applied to porcine skin of different sizes (overlap area). (**D**) The effect of using universal grips (black bars), compared to samples that were appositioned without grips (white bars) on the observed bond strength for each material. All samples contained 60 mol% of phosphoserine, unless otherwise indicated, and were cured for 90 min. Each αTCP, i.e., αTCP2.1, refers to the average particle size (D0.5, surface, in micrometers) of the calcium salt portion of PMC. * and ** indicate *p* < 0.05, and *p* < 0.01, respectively, in comparison to samples with a 1 cm^2^ contact area (**C**), or between grip and no grip samples (**D**). Each data point and group represents a sample size of three (*n* = 3), except in Figure 5A, which contained six samples per group (*n* = 6).

**Figure 6 materials-12-02473-f006:**
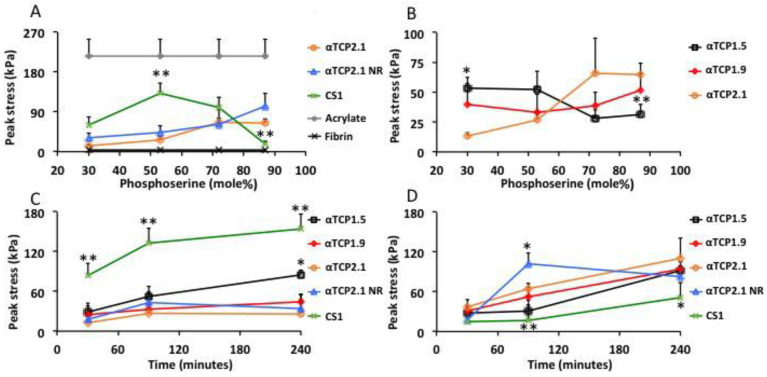
Lap shear strength of PMCs. (**A**) The optimal bond strength of PMCs, with varied ratio of calcium salt to phosphoserine, after curing for 90 min. (**B**) The optimal bond strength of αTCP PMCs of differing particle size, with varied ratios of calcium salt to phosphoserine after curing for 90 min. To ensure that the comparisons in (**A**) and (**B**) represented the true final cure strength, the cure kinetics of each PMC was compared at the optimal mole ratio for (**C**) CS1 PMC (53%) and (**D**) αTCP PMC (87%) after curing for 30, 90, or 240 min. αTCP1.5, αTCP1.9, and αTCP 2.1 refer to the average particle sizes (D0.5, surface, in micrometers) of the calcium salt portion of PMC. * and ** indicate *p* < 0.05 and *p* < 0.01, respectively, with all comparisons made against αTCP 2.1 PMC with respect to each time point or mole ratio. Each data point and group represents a sample size of six (*n* = 6) in Figure 6A–D.

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
