# Peer review of "Adhesive Cements That Bond Soft Tissue Ex Vivo"

_materials, 2019, doi:10.3390/ma12152473_

Reviewer 1 Report

1.      From the Figure 2, the particle size of αTCP1.5 is smaller thanαTCP 2.1. However,  from the SEM image (figure 3), the particle size of αTCP1.5 is larger than αTCP 2.1.  It is inconsistence.

2.      It is necessary to mark the phases corresponding to the XRD peaks on Figure 4.

3.      There should be a space between the number and the unit. Ex: 5 mm not 5mm.

4.      What are the number of sample in Figure 5 and Figure 6?

Reviewer 2 Report

General comments:

This manuscript evaluated phosphoserine modified cements, made of calcium metasilicate and α-tricalcium phosphate, as tissue adhesives for soft tissue applications. The work is very interesting but the experimental section is confusing and not well explained, particularly the samples preparation, and the literature survey is poor.

Some specific comments:

Please move the materials composition obtained through Rietveld refinement from the Section 2.2 to the Results Section. Furthermore, I would not use the words purity and impurity. The synthesis of pure α-TCP is not easy and the presence of β-TCP in the final product is usual, which is attributed to a partial reversion during cooling of the already formed α-TCP.

Please explain the rationale of using fibrin glue as reference.

Please give details about the refined parameters used in the Rietveld refinement in the Section of Materials Characterization.

The preparation of the cements in Section 2.3 is very confusing.

Viscoelastic properties should be performed for the samples.

For SEM images, the particles should be dispersed in water, but with very low concentration. It would be easier to analyze the particles.

The peaks are not identified in figure 4. Please indicate the crystalline phases for each composition.

Author Response

Round  2

Reviewer 1 Report

The authors completely corrected all questions. It is suitable to publish now.

Author Response

No additional comments were provided/required by reviewer #1, only the editor, in this revision.

Reviewer 2 Report

The revised version is considerably improved.

Author Response

No additional comments were provided/required by reviewer #2, only the editor, in this revision.